# Diet and Nutrition in Gynecological Disorders: A Focus on Clinical Studies

**DOI:** 10.3390/nu13061747

**Published:** 2021-05-21

**Authors:** Sadia Afrin, Abdelrahman AlAshqar, Malak El Sabeh, Mariko Miyashita-Ishiwata, Lauren Reschke, Joshua T. Brennan, Amanda Fader, Mostafa A. Borahay

**Affiliations:** 1Department of Gynecology and Obstetrics, Johns Hopkins University School of Medicine, Baltimore, MD 21205, USA; safrin1@jhmi.edu (S.A.); abdashqer@gmail.com (A.A.); melsabe1@jhmi.edu (M.E.S.); mishiwa1@jhmi.edu (M.M.-I.); lreschk1@jhmi.edu (L.R.); jbrenn20@jhmi.edu (J.T.B.); afader1@jhmi.edu (A.F.); 2Department of Obstetrics and Gynecology, Kuwait University, Kuwait City 13110, Kuwait

**Keywords:** diet, nutrition, dietary habits, uterine leiomyoma, endometriosis, polycystic ovary syndrome, gynecological malignancies

## Abstract

A healthy lifestyle and a balanced diet play a paramount role in promoting and maintaining homeostatic functions and preventing an array of chronic and debilitating diseases. Based upon observational and epidemiological investigations, it is clear that nutritional factors and dietary habits play a significant role in gynecological disease development, including uterine leiomyoma, endometriosis, polycystic ovary syndrome, and gynecological malignancies. Diets rich in fruits and vegetables, Mediterranean diets, green tea, vitamin D, and plant-derived natural compounds may have a long-term positive impact on gynecological diseases, while fats, red meat, alcohol, and coffee may contribute to their development. Data regarding the association between dietary habits and gynecological disorders are, at times, conflicting, with potential confounding factors, including food pollutants, reduced physical activity, ethnic background, and environmental factors limiting overall conclusions. This review provides a synopsis of the current clinical data and biological basis of the association between available dietary and nutritional data, along with their impact on the biology and pathophysiology of different gynecological disorders, as well as an outlook on future directions that will guide further investigational research.

## 1. Introduction

Worldwide, women are afflicted by a spectrum of gynecological disorders, ranging from benign entities, such as uterine leiomyoma, endometriosis, and polycystic ovary syndrome, to various gynecological malignancies. These disorders represent a significant source of morbidity by causing bothersome heavy menstruation, debilitating pelvic pain, chronic anovulation, hyperandrogenism, infertility, and even death. Although these conditions are discrepant in nature, they share a common feature: they lack a curative medical treatment that would allow the preservation of functioning reproductive organs. Most medications are only temporarily effective, have undesirable side effects, interfere with pregnancy, and carry a risk of disease recurrence upon discontinuation [1,2,3]. This stresses the importance of fully understanding the pathophysiology of these disorders and the potential risk factors associated with their development and maintenance to mitigate and modify their consequences on health whenever possible.

Diet and health have been among the most complex topics in public discourse and scientific circles. There have long been debates regarding the role that nutritional components and dietary habits play in modulating the risk of gynecological diseases, such as uterine leiomyoma [4,5,6], endometriosis [7,8], polycystic ovary syndrome (PCOS) [9,10], and different gynecological malignancies [11,12,13,14], with the majority of evidence retrieved from epidemiological studies. Evidence suggests that a diet rich in fruit and vegetables, green tea, vitamins, and plant-derived compounds may help prevent gynecological disorders, compared to a diet deficient in vegetables and fruit and high in animal or dietary fats, red meat, and alcohol [6,7,12,14]. Understanding the role of diet in gynecology will change our perspective of how common gynecological diseases develop, progress, and lead to substantial adverse effects on women and will guide us toward pioneering novel diagnostic and therapeutic frameworks to reduce their burden. Most importantly, implementing preventive strategies aimed at changing certain dietary habits may ameliorate the occurrence of a wide array of gynecological diseases. Herein, we reviewed the latest research regarding the role of diet and nutrition in gynecological disorders, with an emphasis on clinical and epidemiological studies.

## 2. Search Criteria

This article provides a comprehensive review of the available literature discussing the role of diet and nutrition in the biological development of various gynecological disorders, emphasizing clinical and epidemiological data. A literature search was conducted using electronic databases, including PubMed of the National Library of Medicine, Google Scholar, Web of Science, and relevant clinical trials addressing diet and nutrition and gynecological disorders. The keywords “diet”, “nutrition”, “fruit”, “vegetables”, “vitamin”, “fat”, “meat”, “fish”, “alcohol”, “coffee”, “tea”, “grain”, “fiber”, “dairy” and “natural compound” combined with “uterine leiomyoma”, “uterine fibroid”, “endometriosis”, “polycystic ovary syndrome”, and “gynecological malignancy” were used. All relevant reports were retrieved, and the corresponding reference lists were systematically searched to identify any additional studies that could be included. Only papers published as full-length articles in English were considered.

## 3. An Overview of Dietary Constituents

### 3.1. Sources of Dietary Elements

Dietary patterns differ in the type of nutritional constituent they contain. Whole grains, such as brown rice, barley, and wheat, are rich in minerals, vitamins, fibers, and phytochemicals, including vitamin E, carotenoids, inulin, and lignans, which play notable roles in modulating immune responses and oxidative stress [15]. Nuts, on the other hand, are rich in healthy (unsaturated) fats, which have been shown to confer a lower risk of obesity, diabetes, and cardiovascular disease. Fruit and vegetables are a known source of vitamins, minerals, and dietary fibers, many of which are notable antioxidants, such as β-carotene, vitamin C, and vitamin E; meat is high in protein and saturated (unhealthy) fatty acids and is a source of N-nitroso compounds, heterocyclic amines, and polycyclic aromatic hydrocarbons, which are known mutagens. Fish, by contrast, provides dietary protein, minerals, and omega-3 fatty acids, which are implicated in reducing the risk of cardiovascular disease [15]. Green tea has been extensively studied for its beneficial effects, and its component epigallocatechin-3-gallate (EGCG), a catechin, has received a great deal of attention for its antioxidant and anti-tumorigenic properties [16]. Later in this review, we will elaborate in depth how different food components and dietary constituents influence the occurrence of common disorders in gynecology.

### 3.2. Role in Health and Disease

Our daily diets consist of a myriad of nutritional constituents that have indispensable roles in maintaining physiological processes by participating in intricate homeostatic cascades. It has been well established that deficiencies in certain nutrients and excess of others can have detrimental consequences by means of disrupting physiology and promoting disease. Among all, micronutrients, consisting of minerals, vitamins, and other trace elements, seem to possess particularly vital roles on cellular grounds. For example, zinc and selenium participate as cofactors in modulating enzymatic activity in hundreds of pathways, whereas zinc has an additional role in gene transcription, depicting their profound effects in regulating cellular functions [17]. Vitamins play an essential role in maintaining health and preventing disease. Tocopherols (vitamin E) and carotenoids (vitamin A) are remarkable in their potential to scavenge reactive oxygen species (ROS), which alongside other zinc- and selenium-containing enzymes, such as glutathione peroxidase and superoxide dismutase, are potent antioxidants. Furthermore, riboflavin and niacin participate in the electron transport chain and cellular energy production, whereas folic acid is involved in DNA synthesis and its deficiency has long been implicated in fetal neural tubal defects in pregnancy [18]. The actions of micronutrients extend to affect the immune system. With regard to the innate immune barrier, iron and zinc are responsible for sustaining epithelial integrity and repair, while calcitriol (vitamin D) promotes a healthier composition of the intestinal microbiota in the gut barrier. On the other hand, vitamins C, E, and B6 enhance T cell differentiation and proliferation and antibody production, emphasizing the prominent role of diet in combating infections [19].

Moving to macronutrients, dietary fats have been shown to have both beneficial and adverse effects on human physiology. On the one hand, trans-fatty acids accentuate inflammation and oxidative stress and influence autophagy and apoptosis. For example, it has been shown that trans-fatty acids potentiate the levels of inflammatory mediators, such as C-reactive protein (CRP) and interleukin (IL) 6, predisposing to atherosclerosis [20]. By contrast, omega-3 fatty acids dampen leukocyte chemotaxis and production of various proinflammatory cytokines, suggesting their use as possible therapeutic agents given their anti-inflammatory properties [21]. It has been found that high glucose levels can initiate several pathophysiologic cascades culminating in drastic outcomes, as already seen in patients with diabetes mellitus. While hyperglycemia mediates vasculopathy, neuropathy, and immunosuppression, it is also implicated in tumorigenesis as will be further discussed in this review. In brief, advanced glycation end products can induce DNA damage, which alongside hyperglycemia-induced ROS, epigenetic modulation, and oncoprotein upregulation, can initiate neoplastic transformation [22]. Similarly, obesity, by means of subclinical inflammation, creates an aberrant adipokine profile that promotes tumorigenesis of benign and malignant nature in women [23].

### 3.3. Role of Diet in Oxidative Stress and Inflammation

Knowing the deleterious effects of oxidative stress and inflammation in the development of various diseases, dietary constituents may have a prominent role in modulating their risk given the antioxidant activity of many. On the other hand, increased intake of certain macronutrients such as carbohydrates can in fact promote oxidative damage. For example, increased caloric intake from high-carbohydrate and high-fat diets dramatically activates the electron transport chain, potentiating superoxide production. Additionally, meat is rich in protein, which if fermented in excess in the gut can produce metabolites, such as ammonia and hydrogen sulfide, that are toxic to the mucosa [15]. When addressing these effects in the context of women’s health, studies have reiterated the effects of diets on various obstetric and gynecological conditions. For instance, Rayman et al. have found that British women with low selenium levels have a higher risk of preeclampsia and pregnancy-induced hypertension. Selenium has been shown to influence eicosanoid synthesis and actions involved in inflammation, thrombosis, immune response, and blood pressure regulation [24]. In settings of oxidative stress, selenium-deficient endothelium loses some of its ability to produce prostacyclin, a vasodilatory eicosanoid, possibly increasing the risk of vascular dysfunction in pregnancy [25]. Similarly, a study has shown that lower copper levels in early pregnancy may increase the risk of pregnancy-induced hypertension, possibly by allowing oxidative stress to progress uninhibited [26]. The role of diet in oxidative stress will be further addressed below in relation to common gynecological diseases.

## 4. Diet and Nutrition in Uterine Leiomyoma

### 4.1. Uterine Leiomyoma

Uterine leiomyomas, also known as uterine fibroids or myomas, are the most common benign tumors of the female reproductive tract, affecting up to 70–80% of women of reproductive age [27], with symptoms ranging from heavy and prolonged menstruation and pelvic pain to subfertility. Although quite prevalent, the precise cellular and molecular culprits of uterine leiomyoma are still unclear [28]. With only few, mostly hormonal, medications that provide short-term relief, surgery (myomectomy or hysterectomy) is the definitive treatment method to date [29]. In fact, uterine leiomyoma is the leading indication for hysterectomy in the United States, with the annual overall leiomyoma-associated costs estimated at $5.9–$34.4 billion [1]. Research over the decades has suggested a number of risk factors for leiomyoma, including age, race, heredity, genetics, epigenetics, sex hormones, endocrine disruptions, obesity, lifestyle factors, and diet [4,5]. Epidemiological evidence suggests that specific dietary components and nutritional factors may be associated with various hormone-related diseases, including uterine leiomyomas (Figure 1) [4,5].

### 4.2. Vegetables and Fruit

Several studies have shown a protective effect of vegetable and fruit intake against uterine leiomyomas, most notably with citrus fruit, apples, cabbage, broccoli, and tomatoes [30,31,32]. In the US Black Women’s Health Cohort Study, the frequency of food items has been determined on the premise of how many servings are consumed per day or month among women before menopause [30]. Four servings of fruit and vegetables each day reduced the risk of uterine leiomyoma in women, compared to just one serving per day. The association was more pronounced among those who consumed several servings of fruit each week, compared to one serving per month [30]. Similarly, a case–control study conducted in premenopausal Chinese women found a protective effect of fruit and vegetable intake against uterine leiomyoma development [33]. It has been suggested that plant-based diets may decrease bioavailable estrogen levels and increase estrogen excretion, thus decreasing the risk of leiomyoma [34]. Another study examined the dietary habits of over 1000 Chinese women using a questionnaire; women with uterine leiomyoma consumed less vegetables compared to healthy women [31]. An Italian case–control study investigated the relationship between green leafy vegetables and fruit and the prevalence of uterine leiomyoma [32]. With weekly frequencies, the risk of surgically confirmed fibroids was inversely associated with intake of green vegetables and fruit, with the exception of carrots, which did not appear to alter this risk [32]. Phytonutrients present in vegetables and fruit may protect against leiomyoma by inhibiting proliferation, inflammation, and fibrosis, inducing apoptosis, and inactivating hormonal or growth factor-related pathways in laboratory studies [35], thus providing a plausible biological basis that may prompt future experiments. A negligible number of dietary nutrients have been examined to date, and more research is needed to clarify the associations in full.

### 4.3. Vitamins

Studies investigating the relationship between vitamins and leiomyoma risk are based on data collected in the Black Women’s Health Study [30]. This study concluded that vitamin A appears to have an inverse association with uterine leiomyoma risk. The results may be attributed to intake of preformed vitamin A derived from animal sources, not from provitamin A derived from fruit and vegetables [30]. However, no correlation was found between leiomyoma and intake of carotenoids, including lycopene [30]. Similar results were obtained in the Nurses’ Health Study II after 10 years of follow-up in a cohort of premenopausal women [36]. The same assessment also showed that intake of β-carotene slightly increased leiomyoma risk, but these effects were restricted to current smokers [36]. In the National Health and Nutrition Examination Survey (NHANES), after adjustments for age, race, education, BMI, and oral contraceptive use, a statistically significant dose–response relationship was identified between serum vitamin A concentrations and uterine leiomyoma, but not with β-carotene [37]. Modern studies reveal no significant association between intake of vitamins C and E and the incidence of leiomyoma [30,37].

In comparison with other vitamins, the value of vitamin D in leiomyoma prevention is significantly stronger. Several studies have examined serum vitamin D concentration in relation to uterine leiomyoma. An analysis of the National Health and Nutrition Examination Survey (NHANES) pointed out that the incidence of leiomyoma is inversely associated with serum vitamin D levels, although specifically among white women [38]. Conversely, a different study concluded that American women aged 35 to 49 with normal levels of vitamin D had an estimated 32% lower risk of leiomyoma occurrence than those who were deficient [39]. However, since the association between serum vitamin D and uterine leiomyoma in both of these studies may be confounded by non-sedentary lifestyles and hence increased sunlight exposure, even after adjusting for BMI, further studies should strongly consider adjusting for such potential confounders to fully elucidate the role of dietary vitamin D intake in leiomyoma risk. Additionally, a statistically significant negative correlation was found between serum vitamin D concentration and leiomyoma in African American women, but not in Caucasian women. This observed association is not limited to leiomyoma occurrence but also related to tumor volume [40]. Similarly, in Indian and Italian studies of women with leiomyoma, vitamin D3 levels were significantly lower in women with leiomyomas compared to those without [41,42]. Notably, a clinical trial in Italy demonstrated that vitamin D supplementation in vitamin D-deficient women with leiomyoma reduced the need for surgery and medical treatments [43]. A number of studies conducted on Turkish, Chinese, and Indian women provided similar results, confirming that vitamin D levels are highly correlated with uterine leiomyoma risk [44,45,46]. An additional study suggested that low 25(OH)D levels may be a risk factor for uterine leiomyoma in individuals with obesity, a positive leiomyoma family history, and a higher level of transforming growth factor 3 [47]. Among studies assessing vitamin D status with uterine leiomyomas, only one study found that vitamin D deficiency and low 25(OH)D serum concentrations did not correlate with uterine leiomyomas in the studied population [38]. Vitamin D is extensively documented throughout the literature for its effects on uterine leiomyoma development and may serve as a potential pharmacological agent for the prevention and treatment of these tumors.

### 4.4. Dietary Fat, Meat, and Fish

There was no correlation between total dietary fat and trans fats and leiomyoma risk in a prospective cohort study of premenopausal African American women after a five-year follow-up [48]. On the other hand, marine omega-3 polyunsaturated fatty acids, namely docosahexaenoic acid, were positively associated with a 49% higher uterine leiomyoma incidence [48]. A Japanese cross-sectional study of premenopausal women found no significant association between dietary fat and uterine leiomyoma [49], which was also shown by Italian and Chinese case–control studies that examined the effect of butter, margarine, eggs, and oil [33]. While the association of dietary fat with uterine leiomyoma remains inconclusive, it is imperative to broadly investigate the role of various fats and fat-containing foods in eliciting a biologically plausible association. Interestingly, there is evidence suggesting that the anti-hyperlipidemic drug class statins may protect against uterine leiomyomas [28,50,51,52], proposing a possible association between leiomyomas and lipid intake and metabolism.

Data regarding the effect of meat and fish consumption on leiomyoma risk are conflicting. Results from an Italian case–control study revealed that significant consumption of meats, such as beef or ham, was associated with an increased risk of leiomyoma [32], but this association was rendered nonsignificant in Chinese populations [33]. Although an inverse association was similarly observed in an Italian case–control study addressing fish consumption and leiomyoma risk [32], other studies found no association between leiomyoma risk and total fish and seafood consumption [33].

### 4.5. Alcohol and Coffee

It is well established that alcohol and coffee intake increases the risk of certain diseases, but the available data regarding uterine leiomyoma risk remain controversial. Several lines of research have shown a link between alcohol consumption and higher leiomyoma risk [49,53]. The Black Women’s Health Study has concluded that leiomyoma is associated with heavier drinking, which may be attributed to phytoestrogens found in beer, as compared to wine or liquor [54]. In the California Teachers Study (CTS), subjects who consumed at least 20 g of alcohol daily had a significantly increased risk of leiomyoma [55] while a case–control study in China did not find such an association [33]. There have been no definitive data regarding whether caffeine consumption increases the risk of uterine leiomyoma in women, except among younger women who consume a high amount of coffee or caffeine (500 mg/day) [54].

### 4.6. Epigallocatechin-3-Gallate (EGCG)

EGCG is a flavonoid found in green tea. A randomized, double-blinded study examined the efficacy of green tea extract in treating women with uterine leiomyoma and improving their quality of life compared to placebo. The findings demonstrated a significant decrease in leiomyoma volume and symptom severity and improvement in health-related quality of life scores [56]. A trial is currently recruiting patients to assess the pharmacokinetics and hepatic safety of EGCG in women with and without uterine fibroids (NCT04177693). A total of 36 women (age groups 18–29 and 30–40) are being given 800 mg EGCG daily alone, EGCG daily with clomiphene citrate, and EGCG daily with letrozole for up to 2 months. Changes in epigallocatechin gallate, epigallocatechin, epicatechin gallate, and 4’-O-methyl-epigallocatechin are listed as primary outcomes whereas changes in direct and total bilirubin, alanine aminotransferase/aspartate aminotransferase, alkaline phosphatase, beta human chorionic gonadotropin (β-hCG), follicle-stimulating hormone (FSH), luteinizing hormone (LH), estrogen, progesterone, anti-mullerian hormone (AMH), and endometrial thickness are listed as secondary outcomes.

### 4.7. Miscellaneous

A number of studies looking at whole grain food intake in Chinese and Italian populations indicated no significant relationship between whole grain intake and leiomyoma prevalence [32,33]. Similarly, there was no significant link between high-fiber diets and leiomyoma development in the Black Women’s Health Cohort study and a Japanese study [30,49].

According to the Italian study, there was no association between milk and cheese intake and uterine leiomyoma risk [32], even though an association was found in a Chinese prospective cohort study [57]. Among African American women, a study found a protective effect of frequent consumption of milk and milk products against leiomyoma occurrence and growth, with no association with butter, cheese, and ice cream, and a minor effect for yogurt consumption [58], warranting the need for further research.

## 5. Diet and Nutrition in Endometriosis

### 5.1. Endometriosis

Endometriosis is a chronic and recurrent disease affecting 6–10% of reproductive-age women and it is a leading cause of pelvic pain and infertility. The formation and growth of endometriotic lesions are dependent on various biological processes that result in the persistence of endometrial-like glandular epithelium and stroma outside the uterine cavity. Endometriosis is an estrogen-dependent disease, and conservative therapeutic strategies are limited to hormonal treatment (combined oral contraceptives, progestins, or gonadotropin-releasing hormone agonists) [59]. Although surgical removal of the endometriotic lesions is an alternate treatment modality, the recurrence rate is up to 50% within five years of surgery. An increasing number of studies have explored diet as a therapeutic strategy for endometriosis, especially in the light of current evidence suggesting an association between benign gynecological conditions, including endometriosis, and cardiometabolic risk and systemic inflammation [23,60]. In the following section, we review the current evidence regarding the association of nutrition and diet with endometriosis, focusing on clinical studies (Figure 2).

### 5.2. Vegetables and Fruit

The risk of endometriosis was found to be inversely related to vegetable and fruit consumption in an Italian population [61]. A significant risk reduction can be seen in women in the highest tertile of intake when compared to women in the lowest tertile, with findings persisting after adjusting for confounding factors [61]. Based on the number of food servings consumed throughout the day, Trabert et al. [62] assessed the role of diets rich in vegetables and fruit in endometriosis, similar to an Italian case–control study. On the contrary, fruit served twice a day was associated with increased risk of endometriosis, but no association was found with vegetables [62].

### 5.3. Vitamins

In a randomized, placebo-controlled clinical trial, 34 women with endometriosis received vitamins C and E or placebo for 6 months. Vitamin C and E supplementation was associated with a decrease in oxidative stress markers in women with endometriosis; however, no improvement in pregnancy rates was noted during or after the intervention [63]. Another study by the same group observed that women with endometriosis had lower intake of vitamins A, C, and E, zinc, and copper, and diminished peripheral oxidative stress markers, compared to women of higher intake [64]. Santanam et al. conducted a clinical trial involving 59 women with pelvic pain and endometriosis or infertility [65]. Patients who received vitamins C and E reported less pain, dysmenorrhea, and dyspareunia, compared to the placebo group. There was a significant decrease in peritoneal fluid inflammatory markers, including the chemokine CCL5/regulated on activation, normal T cell expressed and secreted (RANTES), IL-6, and monocyte chemotactic protein-1 (MCP-1), compared to patients not taking vitamins [65]. These effects may be mediated through the antioxidant properties of vitamins, potentially ameliorating the clinical manifestations of endometriosis.

Moreover, women with endometriosis were noted to have lower serum vitamin D levels compared to women with mild or no endometriosis [66]. There are several clinical trials examining the effect of vitamin D on pelvic pain in endometriosis patients. In randomized, double-blind, placebo-controlled trials, vitamin D supplementation significantly decreased pelvic pain in women with endometriosis [67,68], while no effects were observed by Almassinokiani et al. [69], wherein vitamin D treatment did not significantly reduce dysmenorrhea and/or pelvic pain. Additionally, vitamin D treatment also decreased the total/HDL cholesterol ratio, high-sensitivity CRP, and total antioxidant capacity levels but did not affect other clinical symptoms or metabolic profiles [67]. In another case–control study, women with stage III/IV endometriosis received vitamin D treatment weekly for 12 to 14 weeks, and the expression of β-catenin active form (an important molecule in the Wnt/β-catenin signaling pathway) was significantly reduced after treatment [70].

### 5.4. Dietary Fat, Meat, and Fish

In a large US cohort study, 12 years of prospectively collected data were used to assess dietary fat consumption and endometriosis risk [71]. The results showed that while total dietary fat consumption did not affect endometriosis risk, increased long-chain n-3 fatty acid consumption decreased the risk whereas trans-fat intake increased it [71]. Butter intake was marginally associated among Belgian women with an increased risk of peritoneal endometriosis [72]; however, no association was observed among Italian women [61].

A large Italian study showed that women reporting higher meat consumption had a greater endometriosis risk [61]. In contrast, a study conducted in Belgium showed no significant correlation between endometriosis and meat consumption [72]. Diets rich in red meat seem to correlate modestly with estradiol and estrone sulphate levels, contributing to human circulating steroid hormone concentrations, and thus disease maintenance [73]. However, the Italian and Belgian studies did not reveal any statistically significant association between fish consumption and endometriosis [61,72]. In a meta-analysis by Parazzini et al., endometriosis risk was notably higher among women in the highest tertile of beef and other red meat consumption (OR = 2 [1.4–2.8]) compared to women in the lowest tertile of intake [61].

### 5.5. Whole Grain and Fiber

According to the study by Trabert et al., consumption of whole grains was not associated with endometriosis [62], similar to reports by an Italian case–control study [61]. Alternatively, a study by Savaris and Amaral found a higher fiber intake among women with endometriosis compared to controls, though the sample size was too small to draw any solid conclusions [74]. Despite a higher intake in women with the disease, fiber consumption was still inadequate in this group, which made the authors propose that suboptimal fiber intake might have led to more pronounced inflammation and oxidative stress. Nevertheless, a more biologically plausible mechanism that connects dietary fiber and endometriosis risk needs to be investigated.

### 5.6. Plant-Derived Compounds

#### 5.6.1. Resveratrol

Resveratrol is a phytoalexin polyphenol produced naturally by several plants invaded by bacterial and fungal pathogens. High resveratrol levels are found in grapes, wine, berries, and nuts [75,76]. There are conflicting data concerning resveratrol’s therapeutic effect on endometriosis. A clinical trial included 42 women with endometriosis, of which 16 received oral contraceptives alone and 26 received a combination of oral contraceptives and 30 mg resveratrol. Interestingly, aromatase and COX-2 expressions were reduced in the eutopic endometrium of patients treated with combined oral contraceptives and resveratrol therapy compared to the control group [77]. On the contrary, in another study, 40 mg/day resveratrol with monophasic contraceptives did not improve endometriosis-related symptoms when compared to placebo groups [78].

#### 5.6.2. Epigallocatechin-3-Gallate (EGCG)

In vitro, several groups demonstrated the effect of EGCG on endometriosis [79,80]. The Chinese University of Hong Kong is currently recruiting patients for a randomized, clinical trial on prospective endometrioma patients to study the effects of high-purity EGCG (SUNPHENON EGCgVR; 400 mg, twice per day) versus placebo for 3 months prior to planned surgery. Change in endometriotic lesion size is listed as a primary outcome, whereas pain scores, quality of life, change in lesion growth (by biopsy), change in neovascularization, and monitoring of adverse effects are listed as secondary outcomes (NCT02832271).

#### 5.6.3. Curcumin

Curcumin is a polyphenolic compound found in turmeric, which is derived from the rhizomes of the plant Curcuma longa Linn [81]. The anti-inflammatory and anti-proliferative effects of curcumin on endometriosis were reported previously in in vitro and animal studies [82,83,84,85]. The Vienna-based ENDOFLEX study is currently recruiting patients with endometriosis for an interventional clinical trial randomized for placebo versus the dietary supplement flexofytol, with planned administration of 42 mg of curcumin twice a day for a duration of 4 months. The study defines a possible change in the average pain score, from baseline to 4 months after the onset of treatment, as a primary outcome, and changes in the number of days with pain, alleviation of dyspareunia, dysuria, and dyschezia, as well as changes in quality of life and sexual function as secondary outcomes (NCT04150406).

## 6. Diet and Nutrition in Polycystic Ovary Syndrome

### 6.1. Polycystic Ovary Syndrome (PCOS)

Polycystic ovary syndrome (PCOS) is a very common endocrine disorder in reproductive-aged women, globally affecting 4–21% of this population depending upon the diagnostic criteria applied [86]. This syndrome is characterized by chronic anovulation, hyperandrogenism, and polycystic ovarian morphology, and is associated with increased risks of reproductive disorders, cardiovascular disease, metabolic sequelae, and psychological morbidity [87]. The etiology and pathogenesis of this disease are widely unknown, but a combination of factors including genetic, epigenetic, and environmental influences are thought to play a significant role [88]. Women with PCOS are at increased risk of insulin resistance and associated conditions, including metabolic syndrome, nonalcoholic fatty liver disease, dyslipidemia, hypertension, and obesity-related disorders. Although not required for the diagnosis, the prevalence of insulin resistance has been described in up to 80% of women with PCOS [89], and further research has shown this to occur independently but in an additive manner to obesity [10,89]. While there is no cure for PCOS, first-line management includes conservative treatment through lifestyle interventions that emphasize weight loss and dietary modifications, with efforts to improve insulin sensitivity and prevent long-term health sequelae [9]. In the following section, we provide evidence on the effects of diet and nutrition on PCOS, focusing on clinical studies (Figure 3).

### 6.2. Weight Loss Diets

While no treatment is available for PCOS, research indicates that certain dietary and lifestyle changes can improve the overall metabolic health of women affected by this condition. Modest weight loss, which decreases circulating insulin and androgen concentrations, has been shown to improve the symptoms of PCOS, with the resumption of menstrual cyclicity, spontaneous ovulation and pregnancies, and improvement in overall quality of life [90,91,92,93,94]. However, there is no consensus on which diet is superior for patients with PCOS. Several clinical trials have been conducted to establish which diet is most suitable for PCOS patients, but given the relatively small sample sizes, heterogeneity in dietary interventions, and treatment duration, it is difficult to establish which diet is superior. For the purpose of this review, we discuss randomized controlled trials (RCT) that exclude the use of supplementary medications.

The pulse-based diet, a diet rich in beans, lentils, and chickpeas, was shown to be superior to the therapeutic lifestyle change (TLC) diet, a nutritionally balanced diet in a clinical trial of 95 women enrolled in a 16-week intervention [95]. The pulse-based diet resulted in greater insulin sensitivity, diastolic blood pressure, triglycerides, low-density lipoprotein (LDL) cholesterol, and total cholesterol [95]. Furthermore, in a trial that included 48 patients with PCOS, the dietary approaches to stop hypertension (DASH) diet, a diet rich in fruit, vegetables, and whole grains, and low in fats, cholesterol and sodium, resulted in a significant reduction in insulin resistance markers, waist and hip circumference, total body weight, body mass index (BMI), serum triglycerides, and LDL levels [96]. The benefits of diet were additionally seen in a similar study that included 60 women where the diet reduced serum androstenedione levels, and increased sex hormone-binding globulin (SHBG), thus decreasing total androgens [97].

The effect of calorie distribution per meal has been studied, with the total daily caloric intake standardized to 1800 kcal, but with differences in caloric meal timing distribution: a breakfast diet and a dinner diet, with most of the daily calories consumed during the respective times [98]. The group that consumed most of the calories at breakfast had a significant reduction in glucose and insulin levels, a favorable decrease in testosterone, an increase in SHBG, and improved ovulatory function [98]. These effects were not achieved in the dinner group, suggesting that meal timing and caloric distribution may play a role in women with PCOS [98]. In efforts to analyze the effect of meal frequency on PCOS, a crossover RCT of 40 women with PCOS evaluated the consumption of six meals compared to three meals per day. Findings revealed that subjects with six meals per day resulted in an improved post-intervention oral glucose tolerance test with significant reductions in fasting insulin levels, compared to three meals per day; however, there were no significant differences in hemoglobin A1C and blood lipids [99].

Recent interest in high-protein diets among women with PCOS has been evaluated; however, there is little evidence suggesting this is superior to a standard diet. Moreover, potential concerns regarding aggravation of insulin resistance and impairment of glucose metabolism limit the use of these diets [100,101,102,103,104,105]. In efforts to compare the low-protein, high-carbohydrate diet (LPHC) to a high-protein, low-carbohydrate diet (HPLC), women with PCOS were randomized to either diet. While women in the HPLC diet had a significant reduction in depression and improvement in self-esteem, there was no difference in weight between the groups after adopting the diet for 4 months [106].

In regard to carbohydrate intake, two studies have shown that modest reductions in carbohydrate intake decrease serum insulin, total testosterone, and total cholesterol [107]. The implementation of an isocaloric low-glycemic index diet has also shown promising results, with improvements in insulin sensitivity in women with PCOS. Further epidemiological studies have additionally shown that a low-glycemic index diet reduces the risk of cardiovascular disease, type 2 diabetes, and metabolic syndrome in women; thus the type of carbohydrate intake may have a more significant role in promoting metabolic health than the total amount of carbohydrate intake [108,109,110].

Diets substituting carbohydrates with poly- and monounsaturated fats in obese women with PCOS have been shown to reduce hyperinsulinemia, but no benefit was observed in total cholesterol, triglyceride, and HDL levels [111]. The effects of polyunsaturated fatty acids (PUFA) have further been investigated in women with PCOS. While a PUFA-rich diet resulted in decreased fasting plasma free fatty acids, impaired insulin sensitivity was observed [112]. Additionally, supplementation with PUFA reduced plasma bioavailable testosterone levels; however, there was no significant change in total testosterone, androstenedione, dehydroepiandrosterone sulfate (DHEAS), luteinizing hormone (LH), estrogen, follicle-stimulating hormone (FSH), or SHBG concentrations [113].

Two small studies targeting adolescents with PCOS looked at dietary modifications. No difference between a low-glycemic load and a low-fat diet was observed [114], but both diets resulted in weight loss and improved menstrual regularity. Lastly, a low-glycemic vegan diet was assessed in one small RCT, in which obese women were randomized to either a low-glycemic vegan diet or a low-calorie diet. Though limited by sample size and high attrition rate, obese women randomized to the low-glycemic vegan diet were found to lose significantly more weight after 3 months, but no difference was observed following 6 months of intervention [115].

### 6.3. Vitamin D

A few small studies have looked at the beneficial effects of vitamin D supplementation on PCOS, with conflicting results. Ardabili et al. conducted a clinical trial that included 50 women randomized to receive either three oral treatments of 50,000 IU of vitamin D or three placebo pills over 2 months [116]. There was no difference in fasting serum insulin, glucose levels, or insulin sensitivity between the two groups [116]. Another group randomized women to either receive 4000 IU of vitamin D, 1000 IU of vitamin D, or placebo for 12 weeks [117]. High-dose vitamin D supplementation resulted in significant reductions in total testosterone, free androgen index, hirsutism, and C-reactive protein levels, as well as a significant increase in SHBG and total antioxidant capacity when compared to the low-dose supplementation and placebo groups [117]. The benefits of vitamin D and calcium supplementation were further investigated in two small studies among overweight and obese women with vitamin D deficiency and PCOS, with favorable effects on insulin levels, serum triglycerides, total testosterone, and blood pressure in those with elevated baseline blood pressure [118]. Finally, the effect of weight loss combined with vitamin D supplementation vs. placebo in women with PCOS was studied, with improved menstrual regularity in those subjected to supplementary vitamin D, but no significant difference in weight loss, BMI, or waist and hip circumference was observed [119]. Given the recent evidence that vitamin D plays a role in various metabolic pathways including insulin metabolism and resistance, further research regarding its role in PCOS is warranted [120].

### 6.4. Coenzyme Q10 and Vitamin E

The benefit of coenzyme Q10 (CoQ10), a natural antioxidant, with and without vitamin E has been explored, with supplementation resulting in a favorable cholesterol profile, blood pressure, and fasting blood sugar, reduced total testosterone levels, and an increased SHBG [121,122]. Additionally, in one small study, vitamin E with magnesium supplementation for 12 weeks was found to reduce hirsutism but not significantly alter testosterone or SHBG levels [123].

### 6.5. Inositol

Inositol-phosphoglycan (IPG) has stirred up interest as a potential treatment to improve cellular response to the metabolic secondary messenger pathways following insulin binding with its receptor. IPG has been associated with insulin action, and a deficiency of one of the stereoisomers, D-chiro-inositol, has been linked with insulin resistance [124]. As such, several groups have studied the potential benefit of inositol supplementation in women with PCOS. Forty-two obese patients with PCOS received 2 g of myo-inositol for 8 weeks [125]. Supplementation resulted in a reduction in body mass index and insulin resistance [125]. In patients that had an insulin level above 12 μU/mL, myo-inositol resulted in a significant decrease in fasting insulin levels and the area under the curve of insulin after oral glucose tolerance levels, indicating a greater benefit in obese patients with high fasting insulin levels [125]. The supplementation of a similar isomer, D-chiro-inositol, was studied in a group of overweight and obese women with PCOS patients. Supplementation resulted in an improvement in LH, LH/FSH ratio, insulin response to oral glucose tolerance test, and BMI, and these parameters were improved to a greater extent if the participant had a first-grade diabetic relative [126]. Another study compared the supplementation of myo-inositol to that of D-chiro-inositol for 6 months and found that both isoforms improved ovarian function and metabolism in PCOS patients. Myo-inositol was better at improving the participants’ metabolic profile, while D-chiro-inositol reduced hyperandrogenism [127].

### 6.6. Alternative Therapies

Low selenium levels have been associated in women with PCOS [128]; thus a small RCT was done to test the effect of selenium supplementation in women with PCOS. Selenium supplementation resulted in a higher pregnancy rate, and a decrease in alopecia, hirsutism, and acne. There was no significant difference in testosterone, LH, FSH, nitric oxide, or glutathione (GSH) [129], and further studies are needed.

A study looking at the benefit of soy in women with PCOS showed that a soy diet resulted in a significant decrease in BMI, fasting plasma glucose, total testosterone, insulin resistance, triglycerides, and malondialdehyde. This diet also significantly increased nitric oxide and GSH [130], suggesting that a diet containing soy may offer therapeutic potential for patients with PCOS.

## 7. Diet and Nutrition in Gynecological Malignancies

### 7.1. Gynecological Malignancies

Gynecological malignancies, comprising uterine, ovarian, cervical, vaginal, and vulvar cancers, as well as breast cancer, represent the most common malignant entities in women. In the US alone, more than 90,000 new gynecological cancer cases and about 30,000 deaths were reported in 2017, in addition to 250,000 new cases of breast cancer and 42,000 deaths in the same year [131]. These statistics depict the substantial burden these malignancies impose on women and the economy, necessitating effective preventive measures. Despite their prevalence, the mechanistic aspects of their origins and progression remain largely obscure and are mainly reported as risk factors. A role for diet in modulating the risk of gynecological malignancies has long been described in the literature, with the majority of evidence derived from epidemiological studies. Various micro- and macronutrients, as well as disorders of metabolism and energy expenditure such as obesity, have documented potential roles in influencing these risks [11,12,132]. In the following section, we provide evidence on the effects of diet on women’s cancers, including cervical, ovarian, endometrial, and breast cancers (Figure 4).

### 7.2. Diet and Nutrition in Cervical Cancer

In cervical cancer, the interaction of dietary factors and carcinogenesis has been a matter of controversy, and most data have been conflicting, sparse, and limited by retrospective designs. Studies addressing this association have evaluated the effect of dietary habits on two main outcomes: human papillomavirus (HPV) persistence and squamous intraepithelial lesion (SIL)/invasive squamous carcinoma (ISC) risk [133]. A case–control study in China has concluded that fresh vegetable and green tea intake confers protection against cervical cancer and high SIL, whereas no association was reported for fruit, egg/milk/meat, or soybean intake [134]. This is in contrast to a European study that found a protective effect of fruit intake, but not vegetable intake [12]. The discrepancy in the results may be attributed to varied access of study participants to certain foods in different geographic areas and biases inherent to retrospective data ascertainment [134]. Lending support to these findings, green tea has indeed been shown to halt tumor progression and function synergistically with anti-cancer therapy, whereas fruit and vegetables may dampen cervical cancer risk through ameliorating HPV persistence [135,136,137].

In regard to micronutrient intake, several lines of research have concluded that cervical dysplasia and cancer risk may be significantly reduced in women with higher intake of vitamins C, E, and A, folate, β-carotene, and lycopene [138,139,140]. While experimental studies on the exact mechanisms by which these nutrients exert anti-neoplastic effects in cervical neoplasia are largely lacking, it is biologically plausible to consider their potent antioxidant properties in this context. Women deficient in folic acid, a vitamin necessary for DNA synthesis, are shown to manifest cervical dysplastic changes that are ameliorated upon folate supplementation, implying that folate deficiency may have a role in inducing cervical neoplasia, possibly through defective DNA synthesis and facilitating HPV persistence [141,142]. In addition to questionnaire-based studies, other studies found that low serum folate levels may interact with high-risk HPV to promote cervical intraepithelial neoplasia progression, which may prompt the experimentation of serum folate as a potential biomarker [143]. An excellent systematic review by García-Closas et al. has thoroughly evaluated the state of evidence on micronutrient intake and cervical carcinogenesis [133]. On the other hand, macronutrients, including carbohydrates, proteins, and fat, were not associated with cervical cancer risk [138,144]. When exploring the association of diet with cervical cancer, it is imperative to consider the complex, multifactorial etiology of this cancer as well as the reciprocal effect of latent disease on nutritional status to better elucidate a causal role of diet in this association.

### 7.3. Diet and Nutrition in Ovarian Cancer

Although most evidence on the association between diet and ovarian cancer remains largely inconclusive to date, some studies have investigated an etiological role for diet in this association. In a systematic review of the evidence for dietary intake and ovarian cancer risk, pooled studies indicated a higher risk with increased intake of dietary fat, dairy products, nitrites, and, to a lesser extent, fruit and vitamin C. By contrast, isoflavones, tea, and possibly vegetables may decrease that risk [13]. However, as data are inconsistent due to inherent biases or insufficient sample sizes, it proved difficult to make conclusive remarks about the role of specific dietary constituents compared to others, a notion that was reiterated by the Update Project Report on diet and ovarian cancer risk [145]. In another cross-sectional study conducted in China, animal fat and salted vegetable intake conferred a higher risk of ovarian cancer but vice versa with vegetable and fruit intake [146]. The latter dietary groups may mediate their protective effects through their antioxidant and phytochemical constituents [147,148], whereas salted vegetables may promote carcinogenesis as their nitrite substrates are converted to N-nitroso compounds, a mechanism similar to gastric and esophageal carcinogenesis [149,150].

Of all, most consistent evidence seems to come in favor of fat intake, particularly animal fat, which was shown by both epidemiological and mechanistic studies to increase cancer risk. The National Institutes of Health–American Association of Retired Persons (NIH-AARP) cohort detected a 30% greater risk of ovarian cancer with animal fat intake [151], whereas Larsson et al. showed that dairy products, another source of animal fat, were linked to a 60% higher risk of invasive ovarian cancer [152]. At the cellular level, dietary fat appears to have tumorigenic, inflammatory, and estrogen-elevating properties, which collectively are plausible contributors to ovarian cancer [153,154,155]. Furthermore, the Women’s Health Initiative (WHI) dietary modification trial inferred that low-fat diets followed for >4 years confer a lower risk of ovarian cancer among postmenopausal women [156].

Interestingly, the effect of diet on ovarian cancer appears to occasionally be subtype specific. For example, the Nurses’ Health Study (NHS) and the AARP study found that lactose and total fat are associated with a greater risk of serous ovarian cancer [151,157], whereas a meta-analysis has detected a modest risk reduction for the endometrioid subtype with alcohol intake, as opposed to no association with the epithelial subtype [158]. Regarding obesity, the literature has reported a relatively weak association with ovarian cancer risk, possibly due to retrospective study designs and use of inaccurate measurements of excess body fat [159]. Intriguingly, convincing evidence now shows that increased adult attained height, reflective of genetic, environmental, hormonal, and nutritional factors affecting growth, confers a higher risk of ovarian cancer [145].

### 7.4. Diet and Nutrition in Endometrial Cancer

The association of dietary habits with endometrial cancer has been described in the context of their predisposition to chronic subclinical inflammation and hyperestrogenic states, culprits implicated in endometrial malignancy [160,161]. A recent case–control study done in Italy has concluded that vegetable intake, adherence to a Mediterranean diet, and low dietary inflammatory index confer protection against endometrial cancer [162], lending support to previously documented evidence [163,164]. From a mechanistic perspective, vegetables alter estrogen metabolism, induce antioxidant machinery, and activate the immune system [11], whereas phytoestrogens found in Mediterranean diets may compete with endogenous estrogens [165], possibly antagonizing their effects on the endometrium. On the contrary, animal products can lead to higher dietary inflammatory index and proinflammatory markers such as CRP that can be associated with endometrial cancer [161,166]. While it has been experimentally shown that monounsaturated fatty acid intake may decrease cancer risk through proapoptotic and anti-inflammatory mechanisms [167], clinical observations in this context should be carefully interpreted as they may be confounded by other factors, such as food source and total energy intake [168].

In contrast to ovarian cancer, the relationship between obesity and endometrial cancer is better established. The majority of observational studies, including prospective reports, have identified a significant positive association between body mass index (BMI) and central adiposity parameters, such as waist circumference and waist-to-hip ratio, and endometrial cancer risk [132]. In addition, it was shown that a weight gain of about 20 kg can increase endometrial cancer risk by twofold [132]. Interestingly, early adulthood obesity, and to a lesser extent childhood/adolescent obesity, were associated with a modest increase in endometrial cancer risk, but the results in this context should be interpreted with caution as inaccurate obesity parameters or recall bias might have affected the results [132,169,170]. The biological basis of the association between obesity and endometrial cancer can have multiple explanations. First, excess adiposity augments peripheral conversion of androgens to estrogens by harboring a larger aromatase reserve, especially in postmenopausal women [171], and decreases sex hormone-binding globulin, liberating more bioactive, unopposed estrogen [172]. Second, hyperinsulinemia, commonly co-existing with obesity, may activate mitogenic cascades in the endometrium, where insulin and insulin growth factor 1 (IGF-1) receptors are found to be overexpressed [173]. Third, obesity can create a systemic proinflammatory milieu by means of adipokine secretion, contributing to insulin resistance on the one hand, and to endometrial proliferation on other hand [174,175].

### 7.5. Diet and Nutrition in Breast Cancer

The role of diet in breast cancer has been investigated to a limited extent. Nevertheless, the most compelling evidence was reported for alcohol intake by a substantial number of studies [176,177]. Alcohol consumption is associated with a higher breast cancer risk, especially for hormone-dependent tumors, with a 12% increase in estrogen receptor tumor risk per 10 g ethanol per day [177]. Although unclear and still being investigated, alcohol may mediate this effect indirectly through induction of endogenous sex steroids [178] or directly through ER-dependent or ER-independent signaling pathways [179,180]. Regarding other nutrients, data are more controversial but were documented in the literature, nonetheless. The effect of total fat intake, for example, on breast cancer risk appears to be subtype specific [181]. On the one hand, omega-3 polyunsaturated fatty acids were found to confer a 14% less risk of breast cancer, perhaps through estrogen and adipokine modulating effects [182,183]. Additionally, trans-fatty acids and an increased monounsaturated to saturated fatty acids ratio may heighten a woman’s risk of the tumor [184,185]. In regard to carbohydrates, the results are mixed and vary between pre- and postmenopausal women. Whereas the WHI found no association of dietary glucose load and glycemic index with breast cancer [186], a meta-analysis of observational studies reported a modest positive association between glycemic index and breast cancer risk [187], emphasizing the uncertain nature of this association. Theoretically, a high-glycemic index/glucose load diet can increase insulin release [188], which induces estrogen and IGF-1 and thus cellular proliferation [189,190], providing a plausible mechanism that may prompt future experimental work.

Epidemiological studies have recognized a potentially higher risk of breast cancer among vitamin D-deficient women. A meta-analysis has indeed detected an inverse association between serum vitamin D level and breast cancer risk [191], whereas several other studies failed to re-document it [192], warranting future investigation of this association. When addressing overall dietary patterns, Mediterranean diets comprising fish, vegetables, fruit, legumes, and vegetable oil are associated with a 46% lower risk of breast cancer in one study, whereas more Westernized dietary patterns that include processed meat, high-fat dairy products, sweets, and caloric drinks confer a higher risk of the tumor [14]. Excess adiposity may also signify a higher risk of breast cancer development through dysregulated estrogenic and adipokine profiles [193] and also a higher risk of recurrence and mortality among women with this cancer [194].

## 8. Conclusions and Future Directions

To date, the contribution of diet and nutrition to gynecological disorders remains a largely unexplored avenue that merits substantial future investigation. As most evidence is derived from epidemiological reports, experimental studies that consider potential confounding factors and accurately describe the independent effects of individual nutrients on the development and growth of gynecological diseases should be increasingly conducted. Observational studies addressing this association should shift toward randomized, prospective designs of sufficient power and follow-up periods to detect minor effects. As conservative therapeutic strategies for the prevention and risk reduction of some gynecological disorders are mainly limited to hormonal treatments, exploring the association between these disorders and diet will introduce women to novel therapeutic perspectives. With regard to gynecological malignancies, controlling for various confounding factors may prove important to delineate a precise etiological role for diet. Lastly, randomized controlled trials, wherein specific diets are assigned to different groups, should also be conducted on a wider population basis to better establish the protective or harmful effects of these diets on women’s health.

## Figures and Tables

**Figure 1 nutrients-13-01747-f001:**
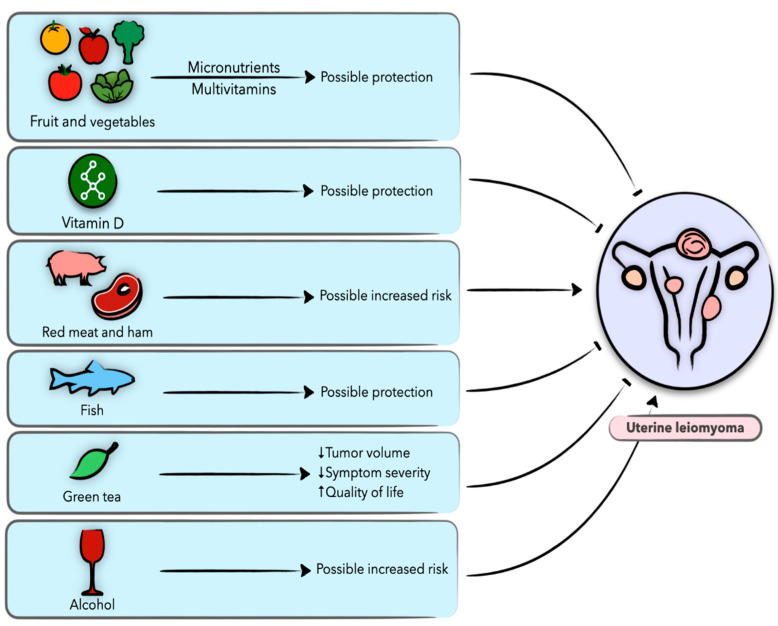
Schematic presentation of the role of diet and nutrition in uterine leiomyoma and the possible underlying biological mechanisms.

**Figure 2 nutrients-13-01747-f002:**
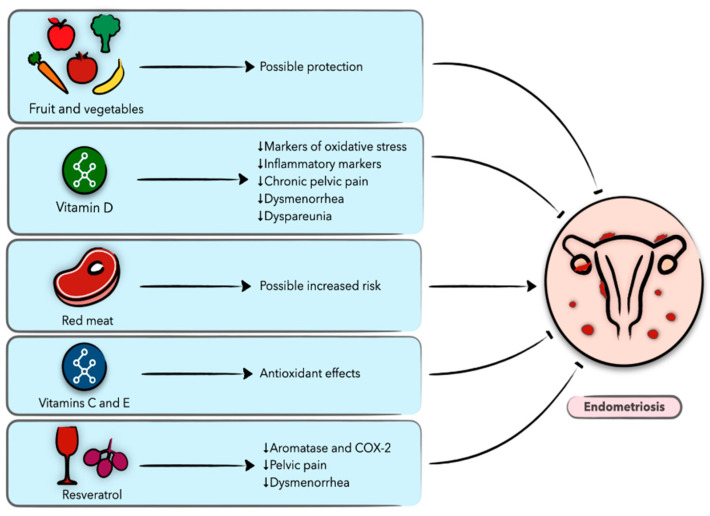
Schematic presentation of the role of diet and nutrition in endometriosis and the possible underlying biological mechanisms.

**Figure 3 nutrients-13-01747-f003:**
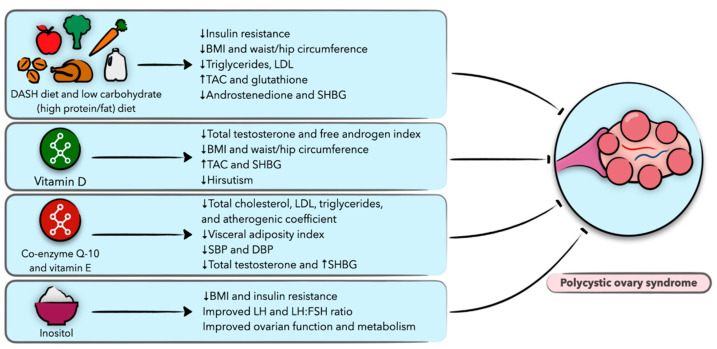
Schematic presentation of the role of diet and nutrition in polycystic ovary syndrome (PCOS) and the possible underlying biological mechanisms.

**Figure 4 nutrients-13-01747-f004:**
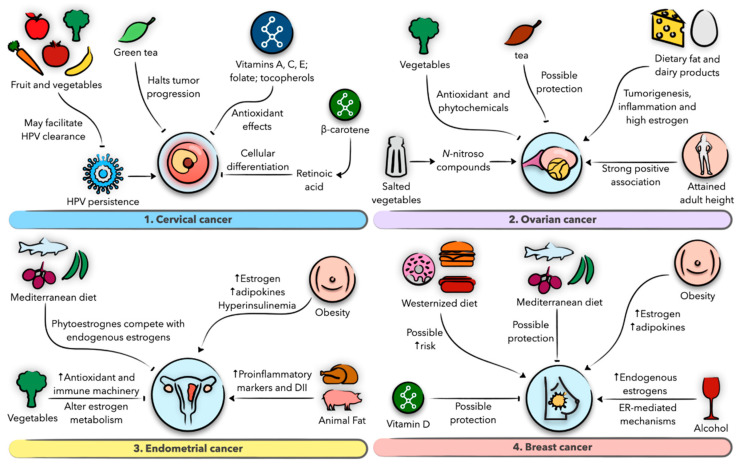
Schematic presentation of the role of diet and nutrition in gynecologic malignancies and the possible underlying biological mechanisms. (**1**) Role of diet in cervical cancer. (**2**) Role of diet in ovarian cancer. (**3**) Role of diet in endometrial cancer. (**4**) Role of diet in breast cancer.

## Data Availability

Not applicable.

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
