# Peer review of "Diet and Nutrition in Gynecological Disorders: A Focus on Clinical Studies"

_nutrients, 2021, doi:10.3390/nu13061747_

Round 1
Reviewer 1 Report
Thank you for the opportunity to review this paper. In this manuscript, Afrin et al. demonstrated the review of the literature discussing the role of diet in the development of various gynecological disorders.
In my opinion, this is an important topic. However, this manuscript should be corrected.
In the introduction, there is no information that will show why the diet affects health or development of diseases. Why is it important to diet study in gynecology ?
Please add one chapter (three sub-sections) (new chapter 3):
3.1- What is the role of different nutrients ? In what physiological processes are involved microelements (selenium, iron, copper, manganese and others) and vitamins (D, C, E, A or B including folic acid). Their deficiency or excess are associated with immune disorders, with increasing inflammation and oxidative stress, as well as affecting DNA and RNA, and others.
How do healthy and unhealthy fats work ?
What is the role of excessive glucose levels ?
What is the role of obesity or underweight in the development of diseases ?
3.2- What is the significance of oxidative stress and inflammation in the development of diseases ?
(in many studies, lower levels of antioxidants, e.g. selenium (but also copper or iron) and vitamins, have been linked to pregnancy complications (e.g Rayman et al. ; Lewandowska et al.)
3.3- What nutrients are included in food groups (e.g. vegetables, fruits, meat, fish/seafood and others).
For example, which food groups contain more antioxidants to protect against oxidative stress (ETC.). What components of green tea can affect health. Why alcohol promotes the development of diseases.
Author Response
- Reviewer 1, Point 1
Thank you for the opportunity to review this paper. In this manuscript, Afrin et al. demonstrated the review of the literature discussing the role of diet in the development of various gynecological disorders.
In my opinion, this is an important topic. However, this manuscript should be corrected.
- Point made by the reviewer: In the introduction, there is no information that will show why the diet affects health or development of diseases. Why is it important to diet study in gynecology?
- Reply to the editor: The authors would like to thank the reviewer for their input. Based on the reviewer’s comments, we further elaborated in the introduction why diet is important in gynecology.
- Specific page and line: Page 3 and lines 52-57.
- Textual change: “Understanding the role of diet in gynecology will change our perspective of how common gynecological diseases develop, progress, and lead to substantial adverse effects on women and will guide us towards pioneering novel diagnostic and therapeutic frameworks to reduce their burden. Most importantly, implementing preventive strategies aimed at changing certain dietary habits may ameliorate the occurrence of a wide array of gynecological diseases.”
- Reviewer 1, Point 2
- Point made by the reviewer: Please add one chapter (three sub-sections) (new chapter 3):
3.1- What is the role of different nutrients? In what physiological processes are involved microelements (selenium, iron, copper, manganese and others) and vitamins (D, C, E, A or B including folic acid). Their deficiency or excess are associated with immune disorders, with increasing inflammation and oxidative stress, as well as affecting DNA and RNA, and others.
How do healthy and unhealthy fats work?
What is the role of excessive glucose levels?
What is the role of obesity or underweight in the development of diseases?
3.2- What is the significance of oxidative stress and inflammation in the development of diseases?
(in many studies, lower levels of antioxidants, e.g. selenium (but also copper or iron) and vitamins, have been linked to pregnancy complications (e.g Rayman et al.; Lewandowska et al.)
3.3- What nutrients are included in food groups (e.g. vegetables, fruits, meat, fish/seafood and others).
For example, which food groups contain more antioxidants to protect against oxidative stress (ETC.). What components of green tea can affect health? Why alcohol promotes the development of diseases.
- Reply to the editor: We would like to thank the reviewer for suggesting this section that will surely enrich our paper. We have added this section with 3 sub-sections to elaborate on sources of dietary elements, role of different dietary constituents in health and disease, and the significance of these constituents in inducing or combating oxidative stress and inflammation. We have also discussed how hyperglycemia and obesity may influence heath and disease. The role of obesity in endometrial cancer was also added in the relevant section.
- Specific page and line: Pages 2-3 and lines 74-151; Page 15 and lines 703-721.

Reviewer 2 Report
It's an interesting study that might be helpful to consider the relationship between nuturtion and gynecologic diseases.
I have several comments;
- Both of the cited papers (26,27) are studies in which serum concentrations of vitamin D are used as explanatory variables. Considering the biosynthetic pathway of vitamin D, the result may reflect sunlight exposure, that is, non-sedentary lifestyle even after adjustment for BMI, as well as dietary intake. Since this paper focuses on the nutritional aspect, an introduction of relevant research on the effects of dietary supplementation alone or the need for further research should be noted.
-
As conflicting research results on meat consumption and endometriosis risk are listed only, it is difficult for readers to draw conclusions. Statistical significance was determined based on the p-value, but there is a problem with the sample size dependency of the p-value itself, so the effect size and its confidence interval of each study should be presented together. If possible, it is best to have a Meta-analysis result. Otherwise the authors had better add the explanation about differrent results of this study from other conflicting sutdies.
-
The correlation between whole grain intake and endometriosis is not well known for its biological plausibility. It is necessary to provide a potential mechanism by which whole grains can influence the development of endometriosis.
- In minor, line no. 726, 798, 817 should be corrected according to the guideline of referrence description.
Author Response
- Reviewer 2, Point 1
It's an interesting study that might be helpful to consider the relationship between nutrition and gynecologic diseases.
I have several comments;
- Point made by the reviewer: Both of the cited papers (26,27) are studies in which serum concentrations of vitamin D are used as explanatory variables. Considering the biosynthetic pathway of vitamin D, the result may reflect sunlight exposure, that is, non-sedentary lifestyle even after adjustment for BMI, as well as dietary intake. Since this paper focuses on the nutritional aspect, an introduction of relevant research on the effects of dietary supplementation alone or the need for further research should be noted.
- Reply to the editor: We thank the reviewer for pointing this out and we made sure to add a statement that reflects the need for further research that accounts for confounders such as sunlight exposure to better establish the relationship between vitamin D and leiomyoma risk.
- Specific page and line: Page 5 and lines 222-226.
- Textual change: “However, since the association between serum vitamin D and uterine leiomyoma in both of these studies may be confounded by non-sedentary lifestyles and hence increased sunlight exposure, even after adjusting for BMI, further studies should strongly consider adjusting for such potential confounders to fully elucidate the role of dietary vitamin D intake on leiomyoma risk.”
- Reviewer 2, Point 2
- Point made by the reviewer: As conflicting research results on meat consumption and endometriosis risk are listed only, it is difficult for readers to draw conclusions. Statistical significance was determined based on the p-value, but there is a problem with the sample size dependency of the p-value itself, so the effect size and its confidence interval of each study should be presented together. If possible, it is best to have a Meta-analysis result. Otherwise the authors had better add the explanation about different results of this study from other conflicting studies.
- Reply to the editor: We thank the reviewer for drawing our attention to this important observation. We added the result of a meta-analysis that assessed the influence of red meat consumption on endometriosis risk.
- Specific page and line: Page 8 and lines 380-383.
- Textual change: “In a meta-analysis by Parazzini et al., endometriosis risk was notably higher among women in the highest tertile of beef and other red meat consumption (OR = 2 [1.4-2.8]) compared to women in the lowest tertile of intake.
- Reviewer 2, Point 3
- Point made by the reviewer: The correlation between whole grain intake and endometriosis is not well known for its biological plausibility. It is necessary to provide a potential mechanism by which whole grains can influence the development of endometriosis.
- Reply to the editor: We thank the reviewer for pointing this out. We have added a statement that explains the possible link of whole grain and fiber intake with endometriosis risk and emphasized the need for further research to come up with more biologically plausible mechanisms.
- Specific page and line: Page 8 and lines 390-394.
- Textual change: “Despite a higher intake in women with the disease, fiber consumption was still inadequate in this group, which made the authors propose that suboptimal fiber intake might have led to more pronounced inflammation and oxidative stress. Nevertheless, a more biologically plausible mechanism that connects dietary fiber and endometriosis risk needs to be investigated.”
- Reviewer 2, Point 4
- Point made by the reviewer: In minor, line no. 726, 798, 817 should be corrected according to the guideline of reference description.
- Reply to the editor: Thank you for pointing this out. The references are now in accordance with the guidelines.

Round 2
Reviewer 1 Report
The authors understood previous comments correctly and they perfectly corrected this manuscript.
I propose to publish this wonderful work.